# Cisplatin Induces Senescent Lung Cancer Cell-Mediated Stemness Induction via GRP78/Akt-Dependent Mechanism

**DOI:** 10.3390/biomedicines10112703

**Published:** 2022-10-26

**Authors:** Nicharat Sriratanasak, Preedakorn Chunhacha, Zin Zin Ei, Pithi Chanvorachote

**Affiliations:** 1Department of Pharmacology and Physiology, Faculty of Pharmaceutical Sciences, Chulalongkorn University, Bangkok 10330, Thailand; 2Center of Excellence in Cancer Cell and Molecular Biology, Faculty of Pharmaceutical Sciences, Chulalongkorn University, Bangkok 10330, Thailand; 3Department of Biochemistry and Microbiology, Faculty of Pharmaceutical Sciences, Chulalongkorn University, Bangkok 10330, Thailand

**Keywords:** stem-like phenotype, chemotherapy, glucose-regulated protein 78, MTJ1, drug resistance

## Abstract

Cellular senescence is linked with chemotherapy resistance. Based on previous studies, GRP78 is a signal transducer in senescent cells. However, the association between GRP78 and stem cell phenotype remains unknown. Cisplatin treatment was clarified to induce cellular senescence leading to stemness induction via GRP78/Akt signal transduction. H460 cells were treated with 5 μM of cisplatin for 6 days to develop senescence. The colony formation assay and cell cycle analysis were performed. SA-β-galactosidase staining indicated senescence. Western blot analysis and RT-PCR were operated. Immunoprecipitation (IP) and immunocytochemistry assays (ICC) were also performed. Colony-forming activity was completely inhibited, and 87.07% of the cell population was arrested in the G2 phase of the cell cycle. mRNA of p21 and p53 increased approximately by 15.91- and 19.32-fold, respectively. The protein level of p21 and p53 was elevated by 9.57- and 5.9-fold, respectively. In addition, the c-Myc protein level was decreased by 0.2-fold when compared with the non-treatment control. Even though, the total of GRP78 protein was downregulated after cisplatin treatment, but the MTJ1 and downstream regulator, p-Akt/Akt ratio were upregulated by approximately 3.38 and 1.44-fold, respectively. GRP78 and MTJ1 were found at the cell surface membrane. Results showed that the GRP78/MTJ1 complex and stemness markers, including CD44, CD133, Nanog, Oct4, and Sox2, were concomitantly increased in senescent cells. MTJ1 anchored GRP78, facilitating the signal transduction of stem-like phenotypes. The strategy that could interrupt the binding between these crucial proteins or inhibit the translocation of GRP78 might beuseful for cancer therapy.

## 1. Introduction

The current standard chemotherapy for lung cancer is cisplatin-based therapy [1]. This chemotherapy can remain in plasma or organs for a decade even when the patients have stopped using it. The retention of the drug causes prolonged DNA damage leading to therapeutic-induced senescence (TIS) at subtoxic doses [2,3,4,5,6].

Cellular senescence is a stable cell cycle arrest which primarily occurs in the G1 and probable G2 phases [7]. These stages inhibit the proliferation and clonogenicity of cancer cells while activating several lysosomal enzymes, including SA-β-galactosidase activity [5]. Previously, it was believed that senescence could improve therapeutic outcomes. However, the aspects of senescent cells have changed. Recent evidence has demonstrated that senescent cells result from DNA damage in chemotherapy, particularly cisplatin which is a conventional chemotherapy, and such cells may be reversible and may escape from the senescence stage [8,9]. Cancer stem cell (CSCs) senescence shows contradictory results. Senescence enforces the self-renewal mechanism and promotes the reprogramming activity of CSCs [10]. Multiple mutations stored in the senescent cells will develop the aggressiveness of cancer cells and lead to relapse [11]. Moreover, a secret senescence-associated secretory phenotype (SASP) secreted from the senescent cells can affect the microenvironment of the surrounding cells and promote a suitable niche for expanding stemness properties [12,13]. Therefore, cellular senescence has both advantages and disadvantages. p53 and p21 play an important role in determining whether damaged DNA cells are senescent or apoptotic. Persistent stimuli cause ataxia telangiectasia mutated (ATM) activation that induces p53 function [14]. p53 upregulates p21 levels, thereby inhibiting cyclin-dependent kinase (CDK) function, and cells undergo cell cycle arrest [14,15].

Glucose-regulated protein 78 (GRP78) belongs to the Hsp70 family, and it is mostly located in the endoplasmic reticulum (ER) or the mitochondria. It plays a role in controlling ER homeostasis. The GRP78 is considered a suppressor of unfolded protein response (UPR) by directly interacting with misfold proteins for degradation [16]. In addition, GRP78 is translocated and anchored at the cell surface membrane by binding to the ER-co-chaperone MTJ1/HTJ1 [17]. After translocation, the GRP78 becomes a signal transducer in various cellular mechanisms, including the PI3K-Akt signaling pathway [18], which is the prominent regulator in the CSC survival pathway [19]. Akt plays a critical role in self-renewal by directly promoting Oct4 and Sox2 upregulation in cancer cells which is associated with tumor initiation and apoptosis resistance [20,21]. In this study, we demonstrated that during cisplatin-induced senescence, GRP78 increasingly accumulates in the cell surface membrane by anchoring with MTJ1/HTJ1 and induces stem-like phenotypes through the Akt signaling pathway. This finding provides insight into the role of GRP78, which might be a potential target to eliminate CSCs. 

Stem-like phenotypes contain pluripotent and self-renewal capacity leading to tumor initiation. Cell surface proteins such as CD44 and CD133 are applied to predict stem-like signatures [22]. Genes associated with self-renewal capacity, such as Nanog, Oct-4, and Sox-2, are overexpressed, thereby upregulating the expression of various types of cell-cycling genes and tumor initiation [23]. However, conventional cytotoxic therapy cannot demolish the CSCs that cause residual disease after treatment, which will be enriched in stem-like phenotype cell populations. These populations can drive various resistance mechanisms and trigger disease relapse [24].

## 2. Materials and Methods

### 2.1. Reagents and Antibodies

Roswell Park Memorial Institute (RPMI) 1640 medium, fetal bovine serum (FBS), antibiotic-antimycotic, L-glutamine supplement, phosphate-buffered saline (PBS), and 0.25% trypsin-EDTA were obtained from Gibco (Grand Island, NY, USA). 3-(4,5-dimethylthiazol-2-yl)-2,5-diphenyltetrazolium bromide (MTT) was acquired from Invitrogen, Thermo Fisher (Waltham, MA, USA). Dimethyl sulfoxide (DMSO), Hoechst 33342, Propidium iodide (PI), and cisplatin were purchased from Sigma Aldrich, Co. (St. Louis, MO, USA). Bovine serum albumin (BSA) and skim milk powder were obtained from Merck Millipore (HES, Germany). An RNeasy kit for the purification of RNA and a Quantinova Reverse Transcription kit were obtained from Qiagen (Hilden, Germany). A Luna^®^ Universal qPCR Master Mix was procured from BioLabs (Hercules, CA, USA). The primary antibodies, c-Myc (#5605), p21 (#2947), p53 (#2527), Akt (#9272), phosphorylated Akt or p-Akt (#4046), Nanog (#4893), Sox-2 (#3579), Oct-4 (#2840), CD44 (#3570), CD133 (#64326), and GAPDH (#5174), and the secondary antibodies for Western blot analysis, anti-rabbit IgG (#7074) and anti-mouse IgG (#7076), were acquired from Cell Signaling Technology (Danvers, MA, USA). The primary antibody, GRP78 (ab21685) and MTJ1 (ab155180), was purchased from Abcam (Cambridge, UK). The secondary antibodies for immunocytochemistry, Alexa Fluor 488 goat anti-rabbit IgG (A11034), Alexa Fluor 488 goat anti-mouse IgG (A11032), and Alexa Fluor 594 goat anti-rabbit IgG (A11037), were obtained from Invitrogen, Thermo Fisher (Waltham, MA, USA). A Senescence β-galactosidase Staining Kit (#9860) was also purchased from Cell Signaling Technology (Danvers, MA, USA). 

### 2.2. Preparation of Cisplatin Stock Solution

Cisplatin was prepared as 5 mM stock suspension in normal saline and stored at −20 °C. Then, the stock suspension was freshly diluted in RPMI completed medium to the required concentrations before treating the cells.

### 2.3. Cell Lines and Culture

Human NSCLC-derived H460 (ATCC^®^ NCI-H460) cells were grown in 10% FBS RPMI with 1% antibiotic-antimycotic. The cells were cultured at 37 °C with 5% carbon dioxide. Cisplatin at 5 µM was added to H460 cells at 30–40% confluency. The cells were continuously maintained with cisplatin at the indicated concentration for 6 days before being used in further experiments.

### 2.4. SA-β-Galactosidase Staining

Cytochemical detection of SA-β-gal was performed with the Senescence SA-β-galactosidase Staining kit (Cell Signaling) according to the manufacturer’s instructions. The treated cells were fixed with a fixation solution for 15 min at room temperature. Then, they were washed with PBS, and the SA-β-gal staining solution at pH 6.0 was added. The cells were incubated at 37 °C with no carbon dioxide overnight. Images of representative fields were captured under ×10 magnification.

### 2.5. Colony Formation Assay

The H460 cells were treated with 5 µM of cisplatin for 6 days before being subjected to form colonies. The living cells were seeded at approximately 300 cells/well and were allowed form colonies for 7 days at 37 °C, 5% carbon dioxide. Then, the colonies were fixed with 4% paraformaldehyde for 15 min at room temperature, strained with crystal violet (0.5% *w*/*v*), and counted using the OpenCFU 3.8 program.

### 2.6. Cell Cycle Analysis

The H460 cells and H460 cisplatin-inducing senescence cells were removed from the well surface by 0.25% trypsin-EDTA. Afterward, the cells were resuspended with serum-free medium and sedimented by centrifuge at 1500 rpm for 5 min. Then, 4% paraformaldehyde was added to fix the cells for 15 min at room temperature. At the end of the incubation time, the cells were washed with PBS and incubated with 20 µg/mL in 0.15 *v*/*v* Triton X-100, 100 µg/mL RNase PBS for 15 min at room temperature in the dark. Subsequently, the cells were analyzed by guavaCyte^TM^ flow cytometry systems (Guavasoft^TM^ Software version 3.3).

### 2.7. RNA Isolation, Reverse Transcription and Quantitative Real-Time PCR (qRT-PCR)

The cells were collected, and total RNA was extracted with the RNeasy kit according to the manufacturer’s instructions. After that, cDNA was synthesized from the extracted RNA using the Quantinova Reverse Transcription kit. Then, cDNA was diluted with RNAse/DNase-free water to a concentration of 10 ng/µL. The thermocycling conditions were as follows: 95 °C for 60 s, 45 cycles at 95 °C for 15 s, 60 °C for 30 s, and 60 °C for plotting the melt curve. The expression levels of each gene were normalized with the GAPDH gene expression level. All samples were performed in triplicate, and the data were calculated using the ΔΔC_t_ method.

### 2.8. Western Blot Analysis

The H460 cells and developed H460 cisplatin resistance cells were collected and incubated with RIPA lysis buffer containing 5 mM Tris-HCl pH 7.6, 150 mM NaCl, 1% NP-40, 1% sodium deoxycholate, and 0.1% SDS for 30 min at 4 °C. Lysates were obtained, and their protein contents were measured using a BCA protein assay kit (Pierce Biotechnology, Rockford, IL, USA). Equivalent amounts of proteins from each sample were separated by SDS-PAGE and transferred to 0.2 µm polyvinylidene difluoride (PVDF) membranes (Bio-Rad). The separating blots were blocked with 5% skim milk in TBST (Tris-buffer saline with 0.1% tween containing 25 mM Tris-HCl pH 7.5, 125 mM NaCl, and 0.1% tween 20) for 2 h and incubated with primary antibody against p21, p53, Akt, p-Akt, c-Myc, GRP78, Nanog, Oct-4, Sox-2, and GAPDH overnight at 4 °C. Afterward, the membranes were incubated with secondary antibody for 2 h at room temperature. The protein bands were exposed using a chemiluminescence substrate and Chemiluminescent ImageQuant LAS4000. The bands were evaluated using Image J software (version 1.52, National Institutes of Health, Bethesda, MD, USA).

### 2.9. Immunocytochemistry Assay (ICC)

The cells were seeded at a concentration of 8000 cells/well and cultured overnight. After that, the supernatant was removed, and the cells were fixed with 4% paraformaldehyde for 15 min at room temperature. They were permeabilized with 0.5% Triton-X 100 in 10% FBS PBS for 5 min, followed by blocking the non-specific protein with 10% FBS in 0.1% Triton-X PBS for 1 h at room temperature. The primary antibodies were diluted to 1:200 *v*/*v* and incubated with the cells at 4 °C overnight. Later, the cells were incubated with secondary antibodies at a concentration of 1:500 *v*/*v* for 1 h before being stained with Hoechst 33342 for 15 min. Glycerol at a concentration of 50% *v*/*v* was added. Then, the images were captured under a fluorescence microscope (Nikon ECLIPSE Ts2). The results were demonstrated in relative mean fluorescence intensity per cell.

### 2.10. Immunoprecipitation (IP)

H460 cells were treated with 5 µM cisplatin for 6 days before being collected and lysed with RIPA buffer. Non-treated cells were used as a control. The magnetic beads from Dynabeads Protein G Immunoprecipitation Kit from Thermo Fisher Scientific Inc. (Waltham, MA, USA) were irrigated with washing buffer and incubated with primary antibody (Ab) GRP78 or MTJ1 in a binding buffer for 10 min. Protein lysate was mixed with the bead–Ab complex at 4 °C overnight. Then, the bead–Ab–antigen complex was washed three times with 200 µL washing buffer. The supernatant was removed, and an elution buffer was added to detach the Ab–antigen complex from the beads. After that, Western blot analysis was performed to detect the MTJ1 or GRP78 protein level, which forms a complex with the pulled-out protein.

### 2.11. Statistical Analysis

The results were demonstrated as the mean ± SEM of at least three independent determinations performed in triplicate. For two-group comparisons, a one-sample *t*-test was calculated using the SPSS software program version 28 (SPSS Inc., Chicago, IL, USA). Statistical significance was considered at *p* < 0.05. GraphPad Prism 5 was used to create graphs for all experiments. Each sample containing at least 1.5 × 10^4^ cells was analyzed. All of the data were calculated based on the results of three replicated samples.

## 3. Results

### 3.1. Cisplatin Induces Senescence in Human Lung Carcinoma Cells

Senescence induction after treatment with a subtoxic concentration of cisplatin was initially evaluated. To investigate the chemoresistance activity of cisplatin in human lung cancer cells, H460 cells were treated with 5 µM of cisplatin for 6 days. The treatment medium was replaced every 3 days, and then SA-β-galactosidase activity was measured using an SA-β-galactosidase staining kit. The treated cells remarkably induced SA-β-gal activity compared with the non-treated control (Figure 1A). A colony formation assay was performed to evaluate whether treated cells in cisplatin-induced senescence lost their proliferative and colony-forming capacity. The results demonstrated that the treated cells were dramatically impaired during colony formation when compared with the non-treated control (Figure 1B,C).

Cell cycle analysis was performed using flow cytometry with propidium iodide staining to confirm whether the senescence activity was induced after cisplatin treatment. The histograms represented that cisplatin notably induced the accumulation of DNA content in the G2/M phase (G2/M control vs. G2/M treatment) in H460 cisplatin-treated cells when compared with the non-treated control, which for 87.07% and 14.47%, respectively (Figure 1D,E). Thus, cisplatin treatment for accounted 6 days led to cell cycle arrest in the G2/M phase.

The senescent program is controlled by multiple interplays of signaling pathways. p53 associated with p21 plays an essential role in inhibiting CDKs [25]. To trigger cell cycle arrest, senescent gene markers, including p53 and p21, were evaluated by qRT-PCR. The primer sequences are shown in Figure 2A. In cisplatin-induced senescent cells, mRNA expression of senescent markers was remarkably increased (Figure 2B). Moreover, the proteins associated with cellular senescence, including c-Myc, p53, and p21, were evaluated. c-Myc is an essential factor in regulating cell cycle progression. The activation of this protein resulted in p21 inhibition [26]. The results showed that the protein expression levels of p53 and p21 dramatically increased, whereas c-Myc protein levels were evidently decreased when compared with the non-treated control (Figure 2C,D).

GRP78 protein expression was decreased in H460 senescent cells. A similar result was reported in a previous study [27]. On the contrary, the expression level of MTJ-1, which is considered a protein associated with GRP78 at the cell surface membrane, was markedly increased when compared with the non-treated control. Moreover, protein expression of the phosphorylated Akt/Akt ratio, which is a downstream regulator of GRP78/MTJ-1, was evaluated. The result showed that the p-Akt/Akt ratio was significantly upregulated in H460 senescent cells (Figure 2E,F).

### 3.2. Senescence Activation Is Related to Mitigating GRP78

The relationship between GRP78 and cellular senescence is not clearly studied; thus, we further demonstrated the location of GRP78 in H460 senescence-induced cells using an immunocytochemistry assay. The cells were also treated with 5 µM of cisplatin for 6 days, and non-treated cells were regarded as the control. The fluorescence intensity of GRP78, which is located at the cell surface membrane, was lower than that located in the ER. However, when the cells were treated with 5 µM of cisplatin for 6 days and when such cells underwent the senescence stage, GRP78 expressed higher intensity at the cell surface membrane than in the ER. The total mean fluorescence intensity of GRP78 in H460 senescence-induced cells was significantly increased compared with the non-treated control (Figure 3A–C). Compared with MTJ1 expression, the results demonstrated the same fluorescence intensity level between the cell surface membrane and ER MTJ1 in H460 non-treated control. However, the fluorescence intensity of the cell surface membrane MTJ1 was notably upregulated in H460 senescence-induced cells. In addition, the mean fluorescence intensity of MTJ1 was markedly increased in H460 senescence-induced cells (Figure 3D–F). We checked the GRP78/MTJ1 complex using co-immunoprecipitation and evaluated the level of the protein complex to confirm its interaction. GRP78 and MTJ1 levels were measured after MTJ1 and GRP78 were removed. Figure 3G–J shows that GRP78 and MTJ1 protein levels were notably elevated in H460 senescence-induced cells, confirming that cellular senescence mediated GRP78/MTJ1 complexation on the cell surface membrane. Moreover, Figure 3K reveals the accumulation of GRP78 expression under higher magnification.

### 3.3. Cellular Senescence Induces Stem-like Phenotype

Self-renewal is an essential characteristic of CSC, and self-renewal transcription factors, including Nanog, Oct-4, and Sox-2, were used to evaluate the mRNA expression by qRT-PCR. Primer sequences such as Nanog, Oct-4, and Sox-2 are presented in Figure 4A. The cells were treated with 5 µM of cisplatin for 6 days to develop senescent cells. The results revealed that all self-renewal markers were remarkably enhanced in cisplatin-treated cells compared to non-treated cells (Figure 4B). The protein expression level of stemness markers, such as CD44, Nanog, Oct-4, and Sox-2, was also evaluated to confirm self-renewal induction. The results indicated that the protein level of stemness markers, CD44, and dominant self-renewal transcription factors, such as Nanog, Oct-4, and Sox-2, were distinctly increased when compared with the non-treated control (Figure 4C,D).

CD133 is indicated as an important biomarker for CSCs. It regulates the Akt signaling pathway leading to apoptosis and chemotherapy resistance [28]. An immunocytochemistry assay was performed to confirm the expression of essential stemness markers, namely, CD133 and CD44. The expression of both CD133 and CD44 was notably elevated compared to the non-treated control (Figure 4E–G).

## 4. Discussion

Cisplatin-based treatment is the first-line traditional chemotherapy for lung cancer disease. Studies have shown that cisplatin is still notably detected in plasma and tissue for decades after the end of the treatment [29]. These subtoxic doses of cisplatin also prolong ATM-dependent DNA damage [30] which finally causes therapy-induced senescence (TIS). This drug escalates the hallmarks of senescence in ovarian cancer cells, nasopharyngeal cancer cells, hepatocellular carcinoma, lymphoma, and lung cancer cells. In the present study, we generated H460 senescence-induced cells by treating them with 5 µM of cisplatin for 6 days and confirmed the senescent phenotype (Figure 1). Ataxia telangiectasia mutation (ATM) was identified as an essential factor triggering cells to escape apoptosis cell death by entering senescence [12]. It contains a serine/threonine protein kinase that is highly sensitive to the signal of DNA double-strand breaks (DSBs) [31]. ATM triggers the cell-responsive machinery of DNA repair and cell cycle via several downstream targets, including p53 [32]. Regarding chemotherapy, the subtoxic concentration of cisplatin which parodies the retention of cisplatin in patients, could prolong DNA damage and persistent DNA damage response (DDR) via the ATM–p53 axis. The activation of ATM could induce senescence and senescence-associated SASP [33]. Moreover, it was demonstrated that GRP78 plays a role in cisplatin-induced senescence through the altered expression of ATM pathway genes such as p53, p21, and cdc2 [34,35].

The activation of p53 leads to cell growth arrest and DNA repair [15]. p21 is a CKD-inhibitory protein (CDKi). The upregulation of this protein causes CDK2 and CDK4 inhibition and induces cellular senescence [36]. p21 is an important protein regulating senescence in a p53-independent pathway [37,38]. Therefore, a subtoxic concentration of cisplatin might induce senescence in two-independent pathways. mRNA and protein expression levels of these ATM pathway genes are dramatically elevated in senescent cells. c-Myc, a CDKi repressor, is also involved [39]. During senescence, c-Myc is strongly decreased (Figure 2). GRP78 is a multifunctional protein which is primarily localized in the ER. This protein is required for the management of UPR. Cellular trafficking of GRP78 to the cell surface membrane under stress conditions can activate the Akt signaling pathway (Figure 2). Several interaction partners of GRP78 are reported, such as α2-macroglobulin, BIK, VDAC, CRIPTO, and MTJ1 [40]. Among these interaction partners, MTJ1 is a transmembrane protein associated with GRP78 translocation to the cell surface. Based on a previous report, silencing MTJ1 can abolish the cell surface expression of GRP78 [41].

MTJ-1 is a transmembrane protein that was demonstrated to be critical for cell surface translocation of GRP78, as knockdown of MTJ-1 could dramatically deplete the cell surface GRP78 [41]. It was also shown that MTJ1 is required for GRP78 catalytic activity and signal transduction [42]. GRP78 function is activated by ATP at the N-terminal domain. The study revealed that MTJ1 (J-domain) binds with the catalytic N-terminal domain of GRP78 [43]. High expression of MTJ1 can convey sufficient GRP78 activation and signaling, leading to cancer aggressiveness and therapeutic drug resistance. This study aims to point out the underlying cause of resistance to cisplatin and demonstrate, for the first time, that the resistant cells with high levels of GRP78/MTJ1 activation might at least be in part related to CSCs.

Resistance to chemotherapies during treatment may be caused by cell adaptation and other means, such as increased drug efflux, drug target modification, survival signal promotion, and stemness [44]. Here, GRP78 is shown to mediate the resistance of lung cancer cells via a UPR concomitant with the stemness of cancer cells. CSCs may share certain characteristics of normal stem cells, including quiescent, enhanced survival, high DNA repair capabilities, and resistance to death stimuli. Recent studies revealed the positive influence of GRP78 on epithelial–mesenchymal transition and CSCs [45,46,47]. In addition, the upregulation of GRP78 in gefitinib-resistant lung cancer cells was shown to associate with increased epithelial-mesenchymal transition and stemness [48]. Other chemotherapeutic drugs, such as doxorubicin and paclitaxel, were shown to induce the unfolded protein response (UPR) regulated by GRP78 and treatment of the cancer cells with these drugs can trigger the translocation of GRP78 to the cell surface [49]. Moreover, oxaliplatin was shown to induce cellular senescence in cancer cells, and the suppression of GRP78 can improve the effectiveness of oxaliplatin treatment [50,51].

This study clearly exhibited the ER localization of GRP78 under normal conditions. By contrast, cytosol expression of GRP78 was remarkably decreased, and the expression levels of GRP78 and MTJ1 at the cell surface were markedly enhanced in H460 senescence-induced cells (Figure 3). GRP78 and MTJ1 coimmunoprecipitated from H460 senescence-induced cells displayed higher interaction of these proteins during senescence (Figure 3). Therefore, GRP78 and MTJ1 complexes are associated with cisplatin-induced senescence. Complexation between these two proteins may not clarify that GRP78 translocate to the membrane, but this protein complex is well known as the form of GRP78 that is induced from ER stress and is ready to switch its function [41,52].

In addition, GRP78 has been discovered on the cell surface of stem cells, and it plays an important role in reprogramming, promoting pluripotent activity, and enriching the stemness genes [53]. It was found that surface GRP78 could promote tumor initiation in CD44+ cells [54]. Remarkably, CD133 and CD44 were dramatically expressed in our established senescence cells (Figure 4). Consequently, this result might support the aggressiveness of the senescence-relapsed cells. The GRP78 surface transducer serves as a regulator of several signaling pathways such as Akt, MAPK, TGF-β, and the hippo signaling pathway [17,54]. Based on previous reports, GRP78 facilitates tumor initiation by eliminating PTEN, an inhibitor of the Akt pathway [55]. The disruption of cell surface GRP78 results in decelerating growth rate and suppression of Akt signaling [52]. Akt signaling has been described as a key regulator for the CSC phenotype by maintaining CSC properties [56]. A series of reports have demonstrated that Akt is directly linked to master pluripotent factors such as Nanog, Oct4, and Sox2 [20,57,58]. Therefore, the stem-like phenotype might be related to GRP78 regulation. The important and common signaling pathway for cell survival and DNA-targeted therapy resistance is Akt. During senescence conditions, the p-Akt/Akt ratio was significantly increased (Figure 2), which could ascribe surface GRP78 as the primary trigger in the Akt signaling pathway. Akt is an upstream regulator of several self-renewal transcription factors [20,21]; thus, we distinguished the mRNA and protein expression of self-renewal transcription factors, namely, Nanog, Oct4, and Sox2. mRNA and protein expression levels of interested self-renewal factors were remarkably increased in senescent cells. This study initially reported that the augmentation of cell surface GRP78/MTJ1 complexes was related to Akt signaling pathway mediated stem-like phenotypes by increasing stemness markers and self-renewal transcription factors. Collectively, GRP78 trafficking occurred during cisplatin-induced senescence, and it was associated with stem-like phenotypic induction leading to drug resistance and cancer relapse in the future. Therefore, targeting the GRP78 protein might prohibit cisplatin-induced senescence and reduce the drug resistance caused by stem-like phenotypic induction. In addition, senescence mediates stem-like phenotypes, which is associated with chemoresistance therapy. Senescence and stemness may be independent events. However, the induction of stem-like phenotype may be observed in cisplatin-resistant cells. A previous study also highlighted that cisplatin treatment could increase CSC by ALDH and CD44 upregulation [59]. Moreover, maintaining cells that circumvent senescence could lead them to acquire stem cell properties [60].

Therefore, our study might be a part of a jigsaw that reveals the cisplatin-induced stem-like phenotype, which may be related to senescence induction. In further verifying the resistance-induced properties, cisplatin-induced senescent cells might be tested with other chemo-drug therapies.

## 5. Conclusions

The sub-toxic concentration of cisplatin which parodies the retention of cisplatin in patients, could alter ATM gene expression and induce cellular senescence. Senescence cells induced from cisplatin treatment demonstrated the upregulation of the GRP78/MTJ1 complex. This protein complex conveys as a form of GRP78, which switch from the major regulator of unfolding protein response to the signaling receptor. This surface transducer could trigger the Akt signaling pathway and regulate stem-like phenotypes. Essential self-renewal transcription factors, such as Nanog, Oct4, and Sox2, were upregulated in the GRP78-mediated senescence mechanism through the Akt signaling pathway (Figure 5). This report would be of benefit to developing a novel treatment strategy.

## Figures and Tables

**Figure 1 biomedicines-10-02703-f001:**
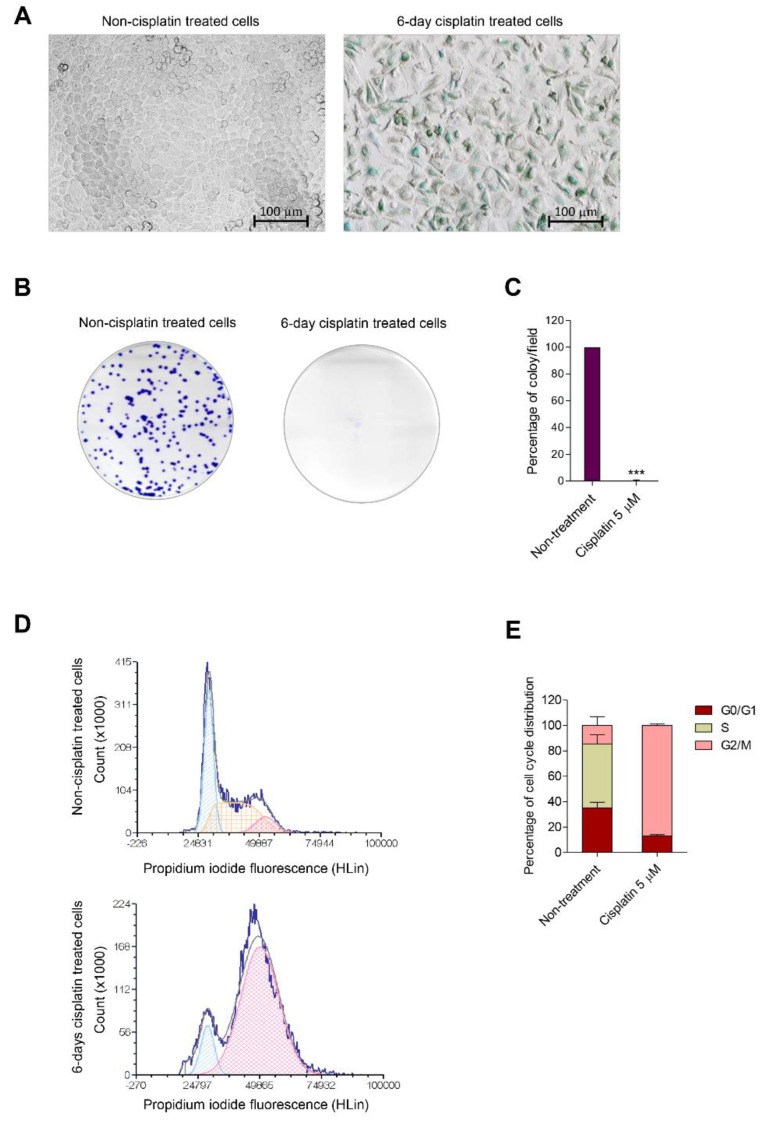
(**A**) The characteristics of H460 senescent cells were evaluated by SA-β-gal staining after continuous treatment with low-dose cisplatin for 6 days compared with the non-treated control. (**B**,**C**) A colony formation assay was performed to evaluate the proliferating inhibition properties of senescent cells after cisplatin treatment for 6 days. The colonies were cultured for 7 days and stained with crystal violet before capturing the image. The percentage of colonies per field was calculated. (**D**) The DNA content in each cell cycle phase was analyzed by flow cytometry with propidium iodide (PI). The histogram represented cell cycle distribution. (**E**) The percentages of DNA content in each phase are presented in a bar graph. Data represent the mean ± SEM (*n* = 3) (*** *p* < 0.001, compared with the non-treated control).

**Figure 2 biomedicines-10-02703-f002:**
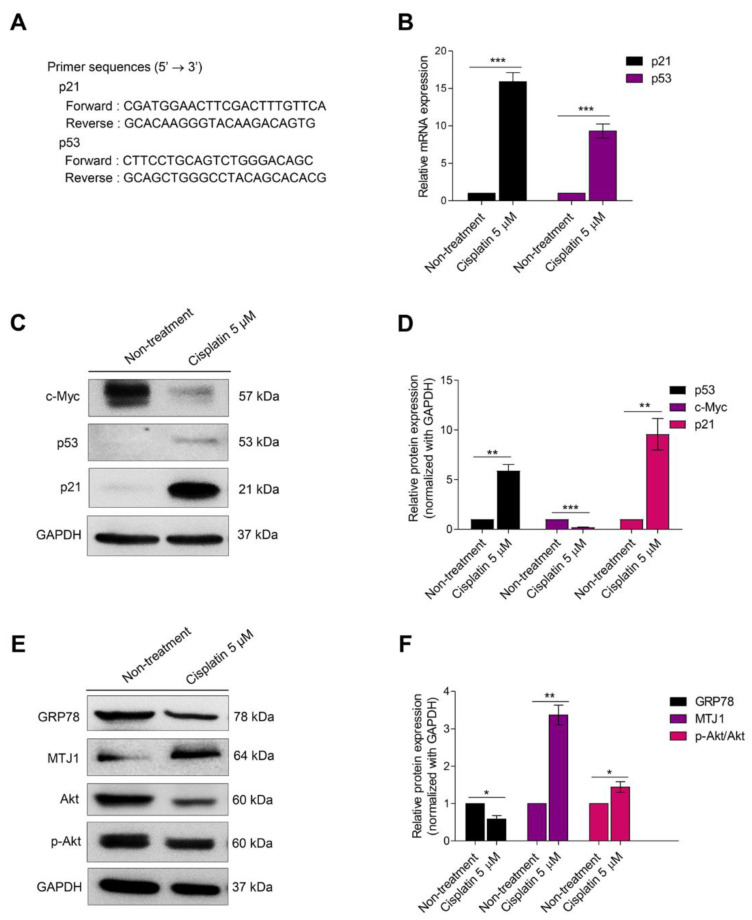
(**A**) The sequences of p53 and p21 for RT-PCR were presented. (**B**) mRNA levels of senescence-related markers, p53 and p21, were examined in H460 and cisplatin-treated cells. Relative mRNA expression levels were calculated compared with non-treated control. (**C**) Western blot analysis was performed to confirm the senescence-related proteins, c-Myc, p53, and p21, in H460 senescent cells when H460 non-treated cells were used as the control. GAPDH was measured to normalize the equal loading of each protein sample. (**D**,**F**) The densitometry of each protein level was demonstrated as a relative protein level. (**E**) To evaluate the GRP78 expression which associated with cellular senescence, MTJ1, which is the essential heterodimer of GRP78 and downstream protein of GRP78, was evaluated by Western blot analysis. The GAPDH protein was examined to confirm equal loading of each protein sample. Data represent the mean ± SEM (*n* = 3) (* 0.01 ≤ *p* < 0.05, ** 0.001 ≤ *p* < 0.01 and *** *p* < 0.001, compared with the non-treated control).

**Figure 3 biomedicines-10-02703-f003:**
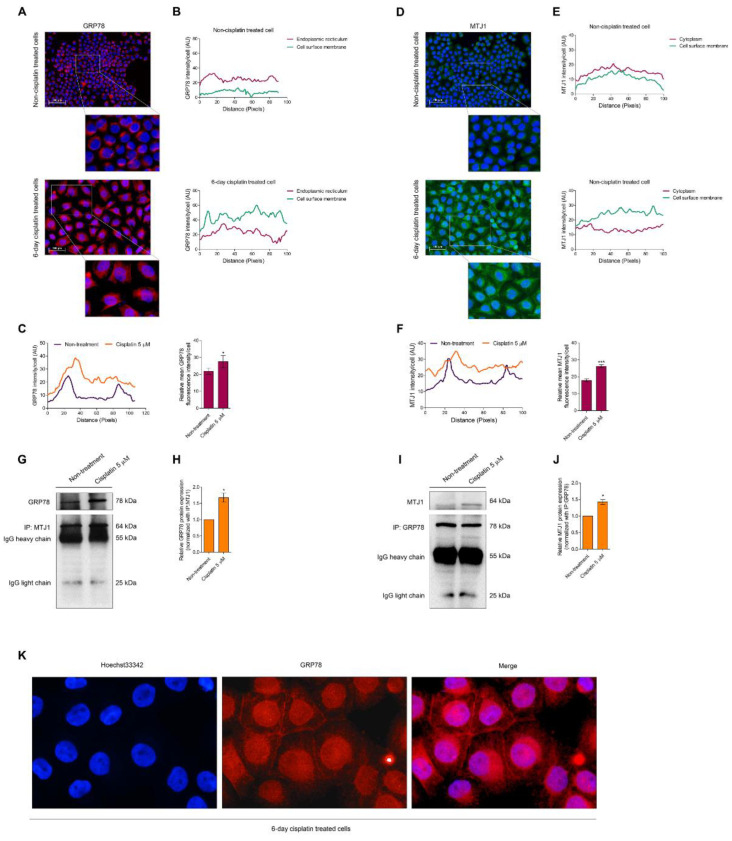
(**A**,**D**) The localization of GRP78 and MTJ1 was evaluated by an ICC assay in H460 non-treated cells and 6-day cisplatin treatment cells. (**B**,**E**) The fluorescent intensity of GRP78 at the nucleus and cell membrane was evaluated by ImageJ software. (**C**,**F**) The fluorescent intensity per cell was also measured. (**G**,**I**) The protein lysates of H460 and H460 senescent cells were collected and incubated with a mixture of beads and GRP78/MTJ1 primary antibodies to extract the protein of interest. Then, GRP78/MTJ1 protein levels were measured by Western blot analysis. (**H**,**J**) The densitometry of each protein level was demonstrated as a relative protein level. The protein levels of each protein were normalized with the protein levels of the pull-out protein. (**K**) The GRP78 protein was captured with 40× magnification. Data represent the mean ± SEM (*n* = 3) (* 0.01 ≤ *p* < 0.05 and *** *p* < 0.001, compared with the non-treated control).

**Figure 4 biomedicines-10-02703-f004:**
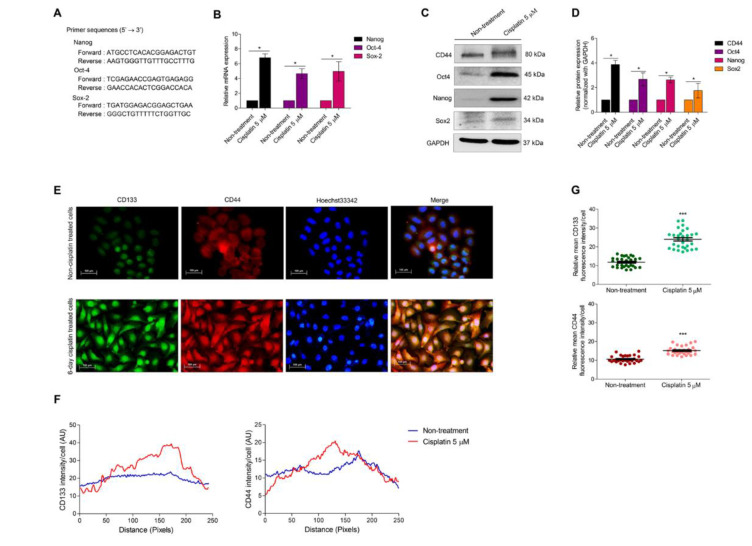
(**A**) The sequences of Nanog, Oct-4, and Sox-2 primers for RT-PCT were presented. (**B**) mRNA levels of pluripotent markers, Nanog, Oct-4, and Sox-2, were examined in H460 and cisplatin-treated cells. Relative mRNA expression levels were calculated and compared with the non-treated control. (**C**,**D**) Western blot analysis was performed to confirm the stemness and pluripotent-related proteins in H460 senescent cells compared with the H460 non-treated control. GAPDH was measured to normalize the equal loading of each protein sample. The densitometry of each protein level was demonstrated as a relative protein level. (**E**) To confirm the inducing stem-like phenotype, the protein expression of CD133 and CD44, which are stemness markers, was evaluated using an ICC assay. (**F**) The fluorescence intensity per cell was measured. (**G**) The relative mean intensity per cell was calculated. Data represent the mean ± SEM (*n* = 3) (* 0.01 ≤ *p* < 0.05 and *** *p* < 0.001, compared with the non-treated control).

**Figure 5 biomedicines-10-02703-f005:**
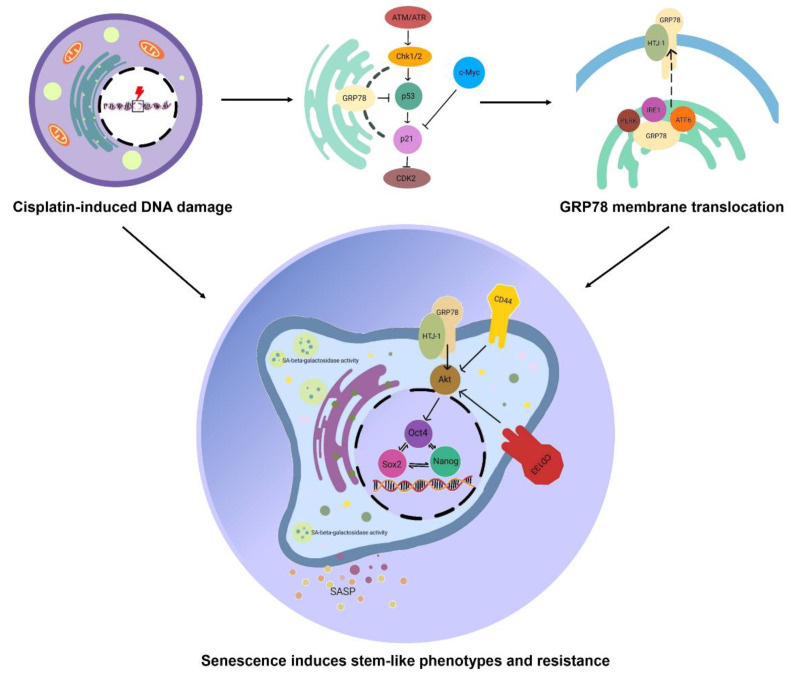
In normal conditions, GRP78 is mainly located in the ER or mitochondria and controls ER homeostasis. It balances the survival of the cells by regulating cell cycle progression through the inhibition of cyclin-dependent kinase inhibitors such as p21. GRP78 interacts with PERK, IRE1, and ATR6 in an unstressed environment and dislocates from these UPR sensors during cellular stress. Cisplatin induces cellular senescence via the DNA damage mechanism via p53 and p21 activation. According to stress stimuli, GRP78 shifts from ER residence to the cell surface membrane, where it is anchored with MTJ1/HTJ1. Signal transduction from cell surface GRP78 can trigger the Akt signaling pathway, resulting in the upregulation of self-renewal transcription factors, Nanog, Oct4, and Sox2. Moreover, the cells which undergo cellular senescence demonstrate higher expression of stemness markers CD133 and CD44.

## Data Availability

Data is contained within the article.

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
