# Peer review of "Cisplatin Induces Senescent Lung Cancer Cell-Mediated Stemness Induction via GRP78/Akt-Dependent Mechanism"

_biomedicines, 2022, doi:10.3390/biomedicines10112703_

Round 1

Reviewer 1 Report

The manuscript entitled "Cisplatin induces senescence lung cancer cells mediated stemness induction via GRP78/Akt-dependent mechanism" is well written and presented.

The authors have published similar work before. However, this manuscript discloses more information like the impact of cisplatin on MTJ1.

The authors need to provide the work's rationale and how it is different from their previous report with detailed explanations. 

Authors should also address the relation between MTJ1 and GRP78.

The discussion part is underdeveloped. The readers may expect more details.

Are there any other drugs also influencing/impacting these membrane proteins?

Author Response

Response: Thank you for your suggestion. This is interesting comment. We would like to clarify that the main novelty of this work is how and by which mechanism, cisplatin regulates cancer stem cell (CSS) of lung cancer. It is clearly known that CSC population is very different from the non-CSC lung cancer cells.
The regulation and link between stemness of cancer cell and cisplatin-mediated senescence are the main finding of this manuscript which is not overlapped with our or other previous studies. As recommended by the reviewer, we have added the discussion as followed; MTJ-1 is a transmembrane protein that was demonstrated to be critical for cell surface translocation of GRP78, as knockdown of MTJ-1 could dramatically deplete the cell surface GRP78 [1] . It
was also shown that the MTJ1 is required for GRP78 catalytic activity and signal transduction [2] . GRP78 function is activated by the ATP at the N-terminal domain. Study revealed that MTJ1 (J-domain) binds with the catalytic N-terminal domain of GRP78 [3] . High expression of MTJ1 can convey the sufficient GRP78 activation and signaling leading to cancer aggressiveness and therapeutic drug resistance. This study aims to point the underlying cause of resistance to cisplatin and demonstrated for the first time that the resistant cells with high level of GRP78/MTJ1 activation might at least in part related to CSCs. Resistance to chemotherapies during therapeutic treatment may cause by cell adaptation and other means like increased drug efflux, drug target modification, survival signal promotion, and stemness [4] . Here, GRP78 is shown to mediate the resistance of lung cancer cells via UPR concomitant with the stemness of cancer cells. CSCs may share certain characteristics of normal stem cells, including
quiescent, enhanced survival, high DNA repair capabilities, and resistance to death stimuli. Recent studies have revealed the positive influence of GRP78 on epithelial-mesenchymal transition and CSCs [5-7] . In addition, the up-regulation of GRP78 in gefitinib-resistant lung cancer cells was shown to associate with increased epithelial-mesenchymal transition and stemness [8] . Other chemotherapeutic drugs such as doxorubicin and paclitaxel were shown to induce the unfolded protein response (UPR) regulated by GRP78 and treatment of the cancer cells with these drugs can trigger the translocation of GRP78 to cell
surface [9] . Moreover, oxaliplatin was shown to induce cellular senescence in cancer cells and the suppression of GRP78 can improve the effectiveness of oxaliplatin treatment [10,11]. 

The answer also inserted into the discussion section.

References:

1. Misra, U.K.; Gonzalez-Gronow, M.; Gawdi, G.; Pizzo, S.V. The role of MTJ-1 in cell surface translocation of GRP78, a receptor for alpha 2-macroglobulin-dependent signaling. J Immunol 2005, 174, 2092-2097, doi:10.4049/jimmunol.174.4.2092.
2. Kaufman, R.J. Orchestrating the unfolded protein response in health and disease. J Clin Invest2002, 110, 1389-1398, doi:10.1172/JCI16886.
3. Chevalier, M.; Rhee, H.; Elguindi, E.C.; Blond, S.Y. Interaction of murine BiP/GRP78 with the DnaJ homologue MTJ1. J Biol Chem 2000, 275, 19620-19627, doi:10.1074/jbc.M001333200.
4. Holohan, C.; Van Schaeybroeck, S.; Longley, D.B.; Johnston, P.G. Cancer drug resistance: an evolving paradigm. Nat Rev Cancer 2013, 13, 714-726, doi:10.1038/nrc3599.
5. Dauer, P.; Sharma, N.S.; Gupta, V.K.; Durden, B.; Hadad, R.; Banerjee, S.; Dudeja, V.; Saluja, A.; Banerjee, S. ER stress sensor, glucose regulatory protein 78 (GRP78) regulates redox status in pancreatic cancer thereby maintaining "stemness". Cell Death Dis 2019, 10, 132, doi:10.1038/s41419-019-1408-5.
6. Klauzinska, M.; Castro, N.P.; Rangel, M.C.; Spike, B.T.; Gray, P.C.; Bertolette, D.; Cuttitta, F.; Salomon, D. The multifaceted role of the embryonic gene Cripto-1 in cancer, stem cells and epithelial-mesenchymal transition. Semin Cancer Biol 2014, 29, 51-58, doi:10.1016/j.semcancer.2014.08.003.
7. Song, J.; Liu, W.; Wang, J.; Hao, J.; Wang, Y.; You, X.; Du, X.; Zhou, Y.; Ben, J.; Zhang, X.; et al. GALNT6 promotes invasion and metastasis of human lung adenocarcinoma cells through O-glycosylating chaperone protein GRP78. Cell Death Dis 2020, 11, 352, doi:10.1038/s41419-020-2537-6.
8. Liao, C.H.; Tzeng, Y.T.; Lai, G.M.; Chang, C.L.; Hu, M.H.; Tsai, W.L.; Liu, Y.R.; Hsia, S.; Chuang, S.E.; Chiou, T.J.; et al. Omega-3 Fatty Acid-Enriched Fish Oil and Selenium Combination Modulates Endoplasmic Reticulum Stress Response Elements and Reverses Acquired Gefitinib Resistance in HCC827 Lung Adenocarcinoma Cells. Mar Drugs 2020, 18, doi:10.3390/md18080399.
9. Raiter, A.; Lipovetsky, J.; Hyman, L.; Mugami, S.; Ben-Zur, T.; Yerushalmi, R. Chemotherapy Controls Metastasis Through Stimulatory Effects on GRP78 and Its Transcription Factor CREB3L1. Front Oncol 2020, 10, 1500, doi:10.3389/fonc.2020.01500.
10. Senescence in Colorectal Cancer Cells Depends on p14(ARF)-Mediated Sustained p53 Activation. Cancers (Basel) 2021, 13, doi:10.3390/cancers13092019.
11. Xi, J.; Chen, Y.; Huang, S.; Cui, F.; Wang, X. Suppression of GRP78 sensitizes human colorectal cancer cells to oxaliplatin by downregulation of CD24. Oncol Lett 2018, 15, 9861-9867,
doi:10.3892/ol.2018.8549.

Reviewer 2 Report

The present report demonstrates sub-toxic concentration of cisplatin could alter ATM gene expression and result in the upregulation of GRP78/MTJ1 complex triggering Akt signaling pathway and regulating stem-like phenotypes. The scientific soundness of the work is unquestionable and well supported by data, however, the overall appeal is lacking. Pt(IV) is being increasingly used, it has better efficacy, and is a lesser-known domain (check out https://pubs.acs.org/doi/full/10.1021/acs.nanolett.2c01850). The author should consider the effect of Pt(IV) along with Pt(II)  (cisplatin) that will really make this work cutting-edge.

Author Response

Response: Thank you for your suggestion. It is very interesting to see the effect of the Pt(IV) as the reviewer have kindly recommended.
For Non-small cells lung cancer (NSCLC) patients who do not have a drug-targetable driver (approximately 85-90%), cisplatin chemotherapy is the recommended first-line treatment [1]. The new information regarding the very new drug (Pt(IV)) can be beneficial, however, considering the clinical relevance and benefit to the patients, we would like to focus on cisplatin as it is the first line drug. Cisplatin, carboplatin, and oxaliplatin (Pt(II)) were approved by the FDA for treatment of NSCLC [1]. However, the resistance mechanism that related to senescence and cancer stem cells remain unclear. Here, we have unraveled the new mechanism of drug resistance related to senescence and stemness of cancer that can be useful for identification of novel targets for drug development as well as the new strategy for reversing resistance. While the platinum (IV) containing compounds can slowly bind to DNA, inhibit DNA
polymerization and reduce resistance [2], there is no Pt(IV)-based drug approved for treatment of lung cancer or NSCLC. Satraplatin (Pt(IV)) is platinum-based drug for the second-line treatment of hormone refractory prostate cancer [3], ovarian cancer [4], and colorectal cancer [5]. As we mainly focus on NSCLC for, and the anti-cancer drugs are very specific to types of cancers, we therefore believe that cisplatin remains the best choice for this research [6,7].
References
1. Fennell, D.A.; Summers, Y.;  Cadranel , J.; Benepal, T.; et. al. Cisplatin in the modernera: The backbone of first-line chemotherapy for non-small cell lung cancer. Cancer Treat Rev 2016, 44:42-50, doi: 10.1016/j.ctrv.2016.01.003.
2. Kasparkova, J.; Novakova, O.; Vrana, O.; Intini, F.; Natile, G.; Brabec. V. Molecular aspects of antitumor effects of a new platinum (IV) drug. Mol. Pharmacol 2006, 70, 1708-1719, doi: 10.1124/mol.106.027730.
3. Armstrong, A.J.; Daniel J George, D.J. Satraplatin in the treatment of hormone-
refractory metastatic prostate cancer. Ther Clin Risk Manag 2007; 3(5): 877-883.
4. Gallerani, E.; Bauer, J.; Hess, D.; Boehm, S.; et.al. A phase I study of the oral
platinum agent satraplatin in sequential combination with capecitabine in the
treatment of patients with advanced solid malignancies. Clinical Trial 2011, 50(7),
105-1110, doi.org/10.3109/0284186X.2010.543697
5. Kalimutho, M.;   Minutolo, A.;  Grelli , S.; Amanda Formosa, A.; et.al. Satraplatin (JM-216) mediates G2/M cell cycle arrest and potentiates apoptosis via multiple death pathways in colorectal cancer cells thus overcoming platinum chemo-resistance. Cancer Chemother Pharmacol 2011, 67(6), 1299-312, doi: 10.1007/s00280-010-1428-4. Epub 2010 Aug 24.
6. Gibson, D. Platinum(IV) anticancer agents; are we en route to the holy grail or to a dead end? J Inorg Biochem 2021, 217, 111353, doi:10.1016/j.jinorgbio.2020.111353
7. Bhargava, A.; Vaishampayan, U.N. Satraplatin: leading the new generation of oral platinum agents. Expert Opin Investig Drugs 2009, 18(11), doi:
10.1517/13543780903362437.

Round 2

Reviewer 2 Report

The reviewer thanks the author for clarification. Please include this as a discussion along with the citations. Once done I recommend this article for publication.